# Antibiofilm Combinatory Strategy: Moxifloxacin-Loaded Nanosystems and Encapsulated *N*-Acetyl-L-Cysteine

**DOI:** 10.3390/pharmaceutics14112294

**Published:** 2022-10-26

**Authors:** Rita M. Pinto, Catarina Leal Seabra, Martine De Jonge, M. Cristina L. Martins, Patrick Van Dijck, Salette Reis, Cláudia Nunes

**Affiliations:** 1LAQV, REQUIMTE, Departamento de Ciências Químicas, Faculdade de Farmácia, Universidade do Porto, 4050-313 Porto, Portugal; 2Laboratory of Molecular Cell Biology, Institute of Botany and Microbiology, KU Leuven, 3001 Leuven, Belgium; 3Instituto de Ciências Biomédicas Abel Salazar, Universidade do Porto, 4050-313 Porto, Portugal; 4i3S, Instituto de Investigação e Inovação em Saúde INEB, Instituto de Engenharia Biomédica, Universidade do Porto, 4200-135 Porto, Portugal

**Keywords:** lipid nanoparticles, bacterial biofilms, combined therapy, biofilm matrix disruption, antibiotic delivery systems

## Abstract

Bacterial biofilms of *Staphylococcus aureus*, formed on implants, have a massive impact on the increasing number of antimicrobial resistance cases. The current treatment for biofilm-associated infections is based on the administration of antibiotics, failing to target the biofilm matrix. This work is focused on the development of multiple lipid nanoparticles (MLNs) encapsulating the antibiotic moxifloxacin (MOX). The nanoparticles were functionalized with d-amino acids to target the biofilm matrix. The produced formulations exhibited a mean hydrodynamic diameter below 300 nm, a low polydispersity index, and high encapsulation efficiency. The nanoparticles exhibited low cytotoxicity towards fibroblasts and low hemolytic activity. To target bacterial cells and the biofilm matrix, MOX-loaded MLNs were combined with a nanosystem encapsulating a matrix-disruptive agent: N-acetyl-L-cysteine (NAC). The nanosystems alone showed a significant reduction of both *S. aureus* biofilm viability and biomass, using the microtiter plate biofilm model. Further, biofilms grown inside polyurethane catheters were used to assess the effect of combining MOX-loaded and NAC-loaded nanosystems on biofilm viability. An increased antibiofilm efficacy was observed when combining the functionalized MOX-loaded MLNs and NAC-loaded nanosystems. Thus, nanosystems as carriers of bactericidal and matrix-disruptive agents are a promising combinatory strategy towards the eradication of *S. aureus* biofilms.

## 1. Introduction

Bacterial biofilms have a massive impact on modern medicine due to the high demand for implantable medical devices. Amongst these, *Staphylococcus aureus* biofilms are the most prevalent and difficult to treat, leading to high rates of morbidity and mortality [1]. When a medical device, such as a catheter, enters the human body, it is prone to bacterial colonization and eventual biofilm formation [2]. Biofilms are three-dimensional structures where an extracellular polymeric substance (EPS) matrix surrounds bacterial cells, providing a protective environment against harsh external conditions, including antibiotics. The EPS matrix plays a key role in the escalating figures of antimicrobial resistance, since it is the first barrier encountered by antibiotics. Due to the difficult diffusion and potential interaction of antibiotics with the EPS matrix components, conventional antimicrobial therapies lack efficiency in eradicating biofilm-associated infections [3,4,5]. Thus, a therapeutic strategy combining antibiotics with matrix-disruptive agents may be the best approach for the eradication of bacterial biofilms, targeting both the EPS matrix and bacterial cells within the structure.

Moxifloxacin (MOX) is a well-known fluoroquinolone with a broad antibacterial spectrum against Gram-positive and Gram-negative bacteria [6]. Despite being a promising antibiotic to treat several bacterial infections, it is not highly efficient against biofilms [7]. Thus, MOX in combination with other antimicrobial agents has been reported for use against bacterial biofilms [8,9]. More recently, a few reports showed the potential of combining MOX with matrix-disruptive agents, such as *N*-acetyl-L-cysteine (NAC) [10]. NAC is a mucolytic agent with antibacterial and antibiofilm properties against several bacterial strains [11,12,13]. Although the antibiofilm mechanism of NAC is still not fully understood, it shows potential in inhibiting biofilm formation and even disrupting mature biofilms [14]. Studies suggest that NAC acts on reducing the production of exopolysaccharides and leads to disruption and degradation of EPS matrix components [12]. However, both MOX and NAC are prone to degradation and clearance upon administration. Both show affinity towards plasma proteins [15,16], and therefore their availability at the biofilm site may be compromised.

Nanotechnology is a powerful tool to improve drug delivery after administration, since nanoparticles protect encapsulated drugs and target them to the biofilm, leading to an increased therapeutic efficacy [17,18]. Amongst several types of nanoparticles, lipid nanoparticles are highlighted due to their high biocompatibility, cost-effectiveness, and controlled drug release [19].

In this work, MOX-loaded multiple lipid nanoparticles (MLNs) were developed to target and eradicate mature *S. aureus* biofilms. The developed nanosystems were functionalized with d-amino acids (d-Phenylalanine, d-Proline, and d-Tyrosine) covalently linked to poly (ethylene glycol) (PEG). PEG was used to increase the circulation time of the nanosystems upon administration, avoiding recognition by the host immune system [20]. The d-amino acids were selected to target and disrupt the biofilms, acting as an adjuvant [21,22]. The produced formulations were physically characterized, and their in vitro cytocompatibility was assessed. The antibiofilm efficacy of the developed MLNs was evaluated alone or in a combinatory therapeutic approach with NAC-loaded nanosystems, which were previously described [10]. In this strategy, NAC would disrupt the biofilm matrix, exposing the dispersed cells to the bactericidal effect of MOX. A schematic overview of the MOX-loaded and NAC-loaded nanosystems used in this work is represented in Figure 1.

## 2. Materials and Methods

### 2.1. Materials

The lipid cetyl palmitate was a kind gift from Gattefossé (Gattefossé, France). DSPE-PEG2000-NH_2_ (1,2-distearoyl-*sn*-glycero-3-phosphoethanolamine-N-[amino(polyethylene glycol)-2000] (ammonium salt)) was purchased from Avanti Polar Lipids Inc. (Alabaster, AL, USA). d-Phenylalanine (Phe), d-Proline (Pro), d-Tyrosine (Tyr), triethylamine (TEA), dicyclohexylcarbodiimide (DCC), N-hydroxysuccinimide (NHS), sodium hydroxide (NaOH), sodium chloride, moxifloxacin hydrochloride (MOX), farnesol 95%, Tween^®^80, Triton^TM^-100x, thiazolyl blue tetrazolium bromide (MTT), D-(+)-glucose monohydrate >99%, Müller-Hinton broth cation adjusted (MHB-Ca^2+^), 2,3-Bis(2-methoxy-4-nitro-5-sulfophenyl)-2H-tetrazolium-5-carboxanilide inner salt (XTT), menadione crystalline, and crystal violet dye were obtained from Sigma-Aldrich (Burlington, MA, USA). Miglyol^®^812 and Span^®^80 were purchased from Acofarma (Barcelona, Spain). *N*-acetyl-L-cysteine (NAC) was purchased from MERK, Millipore Ltd. (Cork, Ireland). The FilmTracer^TM^ LIVE/DEATH^®^ Biofilm Viability Kit was acquired from Thermo Fisher Scientific (Waltham, MA, USA). Chloroform, dimethyl sulfoxide (DMSO), methanol, and acetic acid glacial were obtained from VWR International LLC (Radnor, PA, USA). Dulbecco’s Modified Eagle’s Medium (DMEM), trypsin-EDTA (1×), Penicillin-Streptomycin (PenStrep), Dulbecco’s Phosphate Buffered Saline 10x pH 7.4 (PBS), and Fetal Bovine Serum (FBS) were purchased from Gibco by Life Technologies (Cambridge, MA, USA). L929 cells were acquired from the European Collection of Authenticated Cell Cultures (ECACC) (Salisbury,UK). BD Tryptic soy broth (TSB) and Difco Granulated Agar were acquired from Becton Dickinson (New Jersey, NJ, USA). All components were used without further purification. The *S. aureus* strains ATCC 33591 (methicillin-resistant), ATCC 25923 (methicillin-susceptible), and *S. aureus* strain ATCC 6538 were acquired from ATCC^®^ (Manassas, VA, USA). The bioluminescent strain *Xen36* (Caliper Life Sciences, Hopkinton, MA, USA) was originally derived from *S. aureus* ATCC 49525 and expresses a stable copy of a modified *Photorhabdus luminescens luxABCDE* operon [24,25].

### 2.2. Preparation of MOX-Loaded MLNs

To encapsulate MOX, a new generation of lipid nanoparticles, the MLNs, were produced based on a few modifications from a previously reported protocol [23]. Firstly, the lipid phase constituted by the solid lipid cetyl palmitate (135 mg), the lipid liquid Miglyol^®^812 (305 mg), the lipophilic surfactant Span^®^80 (115 mg), and farnesol (6 mg) was heated at a temperature above the melting point (60 °C) in a water bath. At the same temperature, 5 mg of MOX dissolved in 650 µL of type I water was pre-heated. Unloaded MLNs were prepared by replacing the drug solution with type I water. Then, the aqueous solution was added to the melted lipid phase and sonicated for 3 min at 70% amplitude, using a VCX-130 Vibra-Cell^TM^ sonicator (Sonics & Materials Inc, Newtown, Connecticut, USA), with a CV-18 probe (130 W, 20 kHz). This primary emulsion was left to cool down at room temperature. Further, the hydrophilic surfactant Tween^®^80 (80 mg) was added to the emulsion. d-amino acid conjugates (d-Phe, d-Pro, and d-Tyr) were also added in the functionalized formulations (1 mg each conjugate). The synthesis of these conjugates was previously described elsewhere [10]. The emulsion was then heated for 5 min at 60 °C and sonicated at 80% amplitude for 3 min.

### 2.3. Physical Characterization of MLNs

The hydrodynamic diameter and polydispersity index (PDI) were determined using a Particle Size Analyzer (Brookhaven Instruments Corporation, Software: Particle Sizing v.5 Brookhaven Instruments, New York, NY, USA). The zeta potential of the MLNs was measured using the Zeta Potential Analyzer (ZetaPALS, Brookhaven Instruments Corporation, Software: PALS Zeta Potential Analyzer v.5 Brookhaven Instruments, New York, NY, USA). The system operated with an incidence light angle of 90°, at room temperature. Before the measurements, 5 µL of MLNs was diluted in 8000 µL of type I water.

### 2.4. Determination of the Encapsulation Efficiency and Loading Capacity

The encapsulation efficiency (EE) was calculated by an indirect method to determine the entrapped MOX. In this method, 5 µL of the MOX-loaded MLNs were diluted in 16 mL of type I water. Then, the diluted formulations were transferred to an Amicon Ultra-4 Centrifugal Filter Device 50 kDa (MERCK Millipore, Ltd., Cork, Ireland) and centrifuged at 524× *g* until complete separation of the MLNs from the supernatant (Allegra X-15R Centrifuge, Beckman Coulter, Pasadena, CA, USA). The supernatant was collected to quantify the non-encapsulated MOX using UV-vis spectroscopy (V-660, Jasco Corporation, Software: Spectra Manager V.2, Jasco Corporation, Portland, OR, USA) at 288 nm.

The *EE* was calculated as follows:(1)$$EE\;\left(\%\right)=\frac{Total\;MOX\;amount-unloaded\;MOX\;amount}{Total\;MOX\;amount}\times 100$$

The *LC* was calculated using the *EE*, as follows:(2)$$LC\;\left(\%\right)=\frac{EE\times Total\;MOX\;amount}{Total\;solid\;lipid\;amount}\times 100$$

### 2.5. Evaluation of the Storage Stability

The MLN formulations were stored at room temperature for 24 weeks. The physicochemical storage stability was assessed over time through the measurements of particle size, PDI, zeta potential, *EE*, and *LC*, as described in Section 2.3 and Section 2.4.

### 2.6. In Vitro Cytocompatibility Studies

#### 2.6.1. Cell Viability Assessment

The cell viability of the developed MLNs was evaluated in a fibroblast cell line (L929 cell line), as recommended by the ISO international standard 10993–5:2009 for cytocompatibility assessment studies [26]. These cells were cultured in DMEM supplied with 10% (*v*/*v*) FBS and 1% (*v*/*v*) PenStrep at 37 °C, 5% CO_2_ atmosphere.

The MLN effect on L929 cells was measured using the MTT assay [27,28]. Briefly, the cells were seeded in 96-well plates at a density of 5 × 10^4^ cells/well. When cells reach 80–90% confluence, MLNs with different concentrations of MOX (2.0, 1.0, 0.5, 0.25, and 0.125 µg mL^−1^) were added to the wells and incubated for 24 h, 37 °C, 5% CO_2_. Free MOX was tested at the same concentrations. Triton^TM^-100x (2%, *v*/*v*) and DMEM were used as negative and positive controls, respectively. After the treatment with the MLNs, the medium was replaced by 100 µL of MTT solution (0.5 mg mL^−1^) and incubated for 2 h at 37 °C, 5% CO_2_. Then, the MTT solution was removed, and the formazan crystals were dissolved in 100 µL of DMSO. The absorbance was measured at 550 and 650 nm using a microplate reader (BioTek Instruments Inc., Synergy HT, Software: Gen5 v1.08.4, BioTek Instruments Inc., Winooski, VT, USA). The reading at 650 nm was used to remove the background interference in the measurements.

#### 2.6.2. Hemolysis Activity Assessment

The hemolysis assay was conducted using red blood cells isolated from human blood from healthy blood donors, kindly donated by *Serviço de Hematologia do Centro Hospitalar do Porto.* This procedure was in accordance with the principles of the Declaration of Helsinki. The protocol was adapted from [29]. The samples were centrifuged at 955× *g* for 5 min at 4 °C (Allegra X-15R Centrifuge, Beckman Coulter, Pasadena, CA, USA) to separate the red blood cells from the remaining components of the blood. The cells were washed three times using saline solution 0.85% (*w*/*v*) and further diluted to a volume fraction of 4%. Then, 100 µL of red blood cells were incubated with 100 µL of MLNs for 1 h at 37 °C. The MLNs were tested at a MOX concentration of 512 µg mL^−1^. Free MOX was tested at the same concentration. After the incubation time, the supernatants were collected, and the absorbance of hemoglobin was read at 540 nm and 630 nm using a microplate reader (BioTek Instruments Inc., Cytation, Software: Gen5 v1.08.4, BioTek Instruments Inc., Winooski, VT, USA). The percentage of hemolysis was calculated according to the following:(3)$$hemolysis\;\left(\%\right)=\frac{Abs-Abs\;\left(-\right)}{Abs\;\left(+\right)-Abs\;\left(-\right)}\times 100$$
where *Abs*, *Abs* (−), and *Abs* (+) are the absorbances of the sample, the negative control (saline solution 0.85% (*w*/*v*)), and the positive control (Triton^TM^-100x, 1% (*v*/*v*)), respectively.

### 2.7. In Vitro Antibacterial Studies

The antibacterial activity of both MOX- and NAC-loaded nanosystems against planktonic bacteria was determined using the micro-broth dilution assay for the four strains of *S. aureus* [30]. Briefly, bacteria were grown overnight at 37 °C in TSB and further diluted to a concentration of 2.0 × 10^5^ CFUs mL^−1^ in MHB-Ca^2+^ medium. The bacterial suspension was treated for 24 h at 37 °C with the developed nanosystems in a 96-well round-bottom polypropylene microtiter plate (Greiner Bio One, Kremsmünster, Austria). The produced MLNs and free MOX were tested at MOX concentrations from 0.03 to 2 µg mL^−1^. The unloaded and NAC-loaded LNPs were tested at the solid lipid concentrations of 1 and 2 mg mL^−1^. Free NAC at the concentrations of 0.45 and 0.9 mg mL^−1^ (corresponding to solid lipid concentrations of 1 and 2 mg mL^−1^) was also tested. Untreated bacterial suspension was used as a positive control, and MHB-Ca^2+^ medium was used as a negative control. The MIC was recorded as the lowest concentration that inhibited the visual growth of bacteria. The MBC was further determined by spot-plate (10 µL) in each well with no visual bacterial growth on TSA plates. The plates were incubated at 37 °C overnight, followed by CFU counting. MBC was defined as the lowest concentration resulting in the death of 99.9% of the initial inoculum.

### 2.8. In Vitro Antibiofilm Studies

The potential combinatory effect of the developed MLNs with NAC-loaded LNPs was assessed in *S. aureus* biofilms. For this purpose, NAC-loaded LNPs were produced by the double emulsion method according to the protocol described in [10]. The two nanosystems were studied alone using the microtiter plate biofilm model. The effect on bacterial viability of combining both nanosystems was assessed in a more complex model, the catheter biofilm model.

#### 2.8.1. Microtiter Plate Biofilm Model

Biofilm formation and growth conditions were adapted from the literature [31,32]. Firstly, bacteria were grown overnight at 37 °C on a TSA plate. One or two colonies were collected from the plate and dispersed in TSB 0.6x supplemented with 0.2% (*w*/*v*) glucose to grow overnight at 37 °C, with shaking (240 rpm) (New Brunswick™ Innova^®^ 40/40R, New Brunswick Scientific, Edison, NJ, USA). The initial inoculum was adjusted to an optical density of 0.1 at 600 nm (OD_600nm_) and added to the 96-well polystyrene plates for 90 min, to promote bacterial adhesion. After the incubation time, the non-adhered cells were removed by washing the biofilms twice with 200 µL of PBS 1x. The adhered cells were submerged in 200 µL of fresh medium and further incubated for 24 h at 37 °C to allow the growth of a mature biofilm. The 24-hour-old biofilms were washed twice with PBS 1x and then incubated with the treatment for 24 h at 37 °C. For the treatment with MLNs, the formulations were added at MOX concentrations of 2.0, 1.0, 0.5, 0.25, and 0.125 µg mL^−1^. Free MOX was tested at the same concentrations. For the NAC-loaded LNPs and respective unloaded LNPs, the formulations at the solid lipid concentration of 1 and 2 mg mL^−1^ (corresponding to 0.45 and 0.9 mg mL^−1^ of NAC) or free NAC (0.45 and 0.9 mg mL^−1^) were selected. Untreated biofilms were used as a positive control, and TSB 0.6x supplemented with 0.2% (*w*/*v*) glucose was used as a negative control. This procedure was used for the following assays.

##### Biofilm Biomass Study

Biofilm biomass quantification was assessed through the crystal violet assay based on the reported protocol [33]. Briefly, after the treatment incubation time, the biofilms were washed once with 200 µL of PBS 1x. The biofilms were fixed with 200 µL of 99% (*v*/*v*) methanol for 15 min. After, the methanol was removed, and the wells were left to dry for 15 min. Then, 200 µL of crystal violet solution (1%, *w*/*v*) was added to the biofilms and left to incubate for 5 min. The stained biofilms were washed with type I water to remove the excess of crystal violet solution. To dissolve the remaining stain in the wells, 200 µL of 33% (*v*/*v*) glacial acetic acid was added. Further, the plates were incubated at room temperature for 15 min, with shaking (250 rpm) (VWR Incubating Microplate Shaker, VWR International LLC, Radnor, PA, USA). The absorbance was read at 570 nm using a microplate reader (BioTek Instruments Inc., Synergy HT, Software: Gen5 v1.08.4, BioTek Instruments Inc., Winooski, VT, USA).

##### Biofilm Viability Study

The metabolic activity of the cells within the biofilm was assessed through the XTT reduction assay [34]. After the treatment, the biofilms were washed once with 200 µL of PBS 1x, and 100 µL of XTT solution (1 mg mL^−1^) supplemented with menadione (1 µM) was added in the absence of light. Then, the plates were incubated at 37 °C for 30 min, with shaking (250 rpm) (VWR Incubating Microplate Shaker, VWR International LLC, USA). The absorbance was read at 490 nm using a microplate reader (BioTek Instruments Inc., Synergy HT, Software: Gen5 v1.08.4, BioTek Instruments Inc., Winooski, VT, USA).

#### 2.8.2. Confocal Laser Scanning Microscopy (CLSM) Biofilm Analysis

*S. aureus Xen36* biofilms were formed on µ-Slide 8 Well Glass Bottom (ibidi GmbH, Gräfelfing, Germany) previously washed once with PBS 1x and once with TSB. After, the biofilms were treated for 24 h with the MLN formulations or free drug at the MOX concentration of 0.5 µg mL^−1^. For the treatment with unloaded and NAC-loaded LNPs, the suspensions at a solid lipid concentration of 2 mg mL^−1^ (0.9 mg mL^−1^ of NAC) were used. Free NAC at the same concentration was also tested. Untreated biofilms were used as a positive control. After treatment, the biofilms were washed once with PBS 1x and further stained with the FilmTracer^TM^ LIVE/DEATH^®^ Biofilm Viability Kit (Thermo Fisher Scientific, Waltham, MA, USA) according to the instructions provided by the manufacturer. The stained biofilms were analyzed using the inverted confocal laser scanning microscope Leica Stellaris 8 (Leica Microsystems, Wetzlar, Germany) equipped with the Leica Application Suite X package (LAS X). The excitation/emission wavelengths were set at 488 nm/500–550 nm for Syto9 and 561 nm/570–620 nm for propidium iodide. Images were acquired with a resolution of 1024 × 1024 using a 63X/1.4 oil immersion objective. An area of ~180 µm (X) × 160 µm (Y) was screened in ~1 µm Z-intervals (Z-stack) for 20 µm.

#### 2.8.3. Catheter Biofilm Model

In the in vitro catheter biofilm model, the biofilms were formed inside triple-lumen polyurethane central venous catheters (Certofix duo/trio; B. Braun Melsungen AG, Germany), based on a protocol described in the literature [31]. Firstly, 1 cm long catheter pieces were incubated with 100% (*v*/*v*) FBS overnight at 37 °C. The bacterial inoculum was adjusted as described in Section 2.8.1. Then, the catheter pieces were incubated with 1 mL of bacterial inoculum for 90 min at 37 °C to promote adhesion of bacterial cells to the material. After the adhesion period, the biofilms were washed twice with 1 mL of PBS 1x. The adhered cells were submerged in 1 mL of fresh TSB 0.6x supplemented with 0.2% (*w*/*v*) glucose and further incubated for 24 h at 37 °C. The mature biofilms were then treated with MLNs and NAC-loaded LNPs, alone or in combination, for 24 h at 37 °C. The MLNs were tested at a final MOX concentration of 0.5 µg mL^−1^, while NAC-loaded LNPs were added to a final NAC concentration of 0.9 mg mL^−1^. The unloaded formulations and the free compounds were also tested at the corresponding concentrations. Untreated biofilms were used as a positive control, and TSB 0.6x supplemented with 0.2% (*w*/*v*) glucose was used as a negative control. Catheter pieces were then washed twice with 1 mL of PBS 1x and sonicated (Branson 5510 Ultrasonics bath, Marshall Scientific, Hampton, NH, USA) for 10 min in 200 µL of PBS 1x to disperse the biofilms. The samples were serially diluted in PBS 1x and spot-plated (10 µL) on TSA plates. The plates were incubated overnight at 37 °C, followed by CFU counting.

### 2.9. Statistical Analysis

Statistical analysis was performed using Graphpad Prism Software (version 7.03; IBM, New York, NK, USA). Data are expressed as mean ± SD, and each assay was performed at least three independent times, except for CLSM biofilm analysis, which was repeated two times.

## 3. Results

### 3.1. Physical Characterization of MLNs and Storage Stability

The developed MLNs were characterized regarding their mean hydrodynamic diameter, PDI, zeta potential, EE, and LC, over a storage period of 24 weeks, at room temperature (Figure 2). Both the encapsulation of MOX and the functionalization of the nanoparticles led to an increase in the hydrodynamic diameter compared to the unloaded MLNs (uMLNs) (Figure 2A). This outcome may be caused by steric effects inside the lipid matrix and at the surface of the nanoparticles promoted by the drug and the conjugates, respectively. Nevertheless, all formulations showed a mean hydrodynamic diameter below 300 nm, with no significant changes observed in the hydrodynamic size of the formulations over 12 weeks of study. After 24 weeks of storage, significant changes in the hydrodynamic diameter were observed for uMLNs, MOX-MLNs, and F-uMLNs. The low PDI (<0.15) observed for all formulations suggests that they were monodisperse in size and did not tend to form aggregates over time, at the defined storage conditions [35]. This low tendency to aggregate and the high stability of the nanoparticles in suspension were also confirmed by their low zeta potential (below −30 mV) (Figure 2B) [36].

The EE and the LC of the MLNs were also studied during the 24 weeks of study (Figure 2C,D). For 12 weeks, the EE for both non-functionalized and functionalized MOX-loaded MLNs was satisfactorily high (above 75%), revealing no significant leakage of the drug over this time. The LC values were also unaffected during this time. Interestingly, F-MOX-MLNs showed a significantly higher EE and LC compared to MOX-MLNs, which suggests that the surface functionalization improves the entrapment of MOX. It is hypothesized that this increase in EE and LC may be a consequence of the increased steric effects promoted by the functionalization at the nanoparticles’ surface. After 24 weeks, the EE and LC of both MOX-MLNs and F-MOX-MLNs significantly decreased, revealing a significant drug release. Hence, the developed formulations were stable under storage conditions for 12 weeks.

Overall, the physical characteristics of the developed MLNs are crucial for a safe in vivo systemic administration. It is reported that particles entering the bloodstream should be significantly smaller than the diameter of human capillaries (around 5 µm) and with no tendency to form aggregates [36]. Besides particle size, the surface charge may also play a role in the biodistribution and pharmacokinetics of the nanoparticles. Negatively charged nanoparticles have prolonged circulation half-lives as compared to highly positively charged ones [36]. In addition, positively charged nanoparticles are considered more cytotoxic to human cells, due to a higher cellular uptake [37]. Considering these properties and their influence on the in vivo fate of the particles, the produced formulations are suitable for systemic administration.

### 3.2. In Vitro Cytocompatibility Studies

The cytocompatibility of the developed MLNs was assessed using a fibroblast cell line (L929) and human red blood cells. The cell viability of fibroblasts after a 24 h treatment with both unloaded and MOX-loaded MLNs was not significantly reduced at any of the concentrations tested (Figure 3A). However, a significant decrease in cell viability was observed for free MOX at 0.25 µg mL^−1^ and higher concentrations, compared with the untreated cells. At the highest concentrations of MOX (1 and 2 µg mL^−1^), free MOX revealed a potential toxicity effect, with cell viability values below 70% [26].

To evaluate the potential effects of the developed MLN suspensions on red blood cells after a systemic administration, a hemolysis assay was performed (Figure 3B). For this assay, the MLNs and free MOX were tested at a MOX concentration of 512 µg mL^−1^. At this concentration, both the formulations and free drug showed a hemolytic activity below 2.5%. According to the ASTM E2524-08 standard, a hemolysis percentage below 5% does not cause damage to the red blood cells. Since the developed formulations meet this criterion, they are a potentially safe nanosystem for systemic administration.

From the cytocompatibility studies in both fibroblasts and human red blood cells, it is possible to conclude that the developed MLN suspensions present a safe profile for further in vivo applications.

### 3.3. In Vitro Antibacterial Studies

Prior to in vitro antibiofilm studies, both MOX- and NAC-loaded nanosystems were tested for their antibacterial activity against planktonic bacteria. The efficacy of the developed MOX-loaded nanosystems against planktonic bacterial cells can be found in the Appendix A. At the highest tested concentration, unloaded MLNs showed no inhibition of planktonic bacterial growth for any of the tested strains. The MOX-loaded formulations (MOX-MLNs and F-MOX-MLNs) showed MIC values higher than free MOX. This outcome suggests that not all encapsulated MOX was released from the nanoparticles during the treatment period. The MIC values obtained for both MOX-loaded formulations and the free antibiotic were below the MIC breakpoint reported by EUCAST (0.25 mg L^−1^) [38].

For the NAC-loaded formulations and corresponding unloaded formulations, no bacterial inhibition was found at the highest concentration tested, 0.9 mg mL^−1^ of NAC (data not shown). These results were expected since Drago et al. (2013) reported MIC values of NAC for *S. aureus* strains ranging from 12 to 24 mg mL^−1^ [39].

### 3.4. In Vitro Antibiofilm Studies

#### 3.4.1. Biofilm Viability and Biomass Studies

The in vitro antibiofilm efficacy of the developed MLNs was initially assessed by quantification of the biofilm viability (Figure 4A,C) and biomass (Figure 4D,F), using the microtiter plate biofilm model for three *S. aureus* strains ATCC 33591 (methicillin-resistant), ATCC 6538 (susceptible), and *Xen36* (bioluminescent strain). In addition, the antibiofilm activity against the *S. aureus* ATCC 25923 (susceptible) was also assessed (Appendix A.

For all tested strains, it is possible to observe that MOX-loaded MLNs and free MOX had a higher effect on the biofilm viability compared to the unloaded MLNs (Figure 4A,C). uMLNs did not show any significant reduction of bacterial viability for both MRSA ATCC 33591 and *S. aureus* ATCC 6538 strains at any tested concentration. For the *Xen36* strain, this formulation only showed a significant reduction of viability at 0.5 µg mL^−1^ and higher concentrations. However, the functionalized unloaded MLNs (F-uMLNs) showed a significant reduction of *Xen36* biofilms even at 0.125 µg mL^−1^. The F-uMLNs also showed a significant effect on MRSA biofilms at 0.25 µg mL^−1^ and higher concentrations. These results suggest that the functionalization of the MLNs with d-amino acids (Phe, Pro, and Tyr) may have an impact on the bacterial viability of *S. aureus* strains. The effect of a mixture of these three d-amino acids against *S. aureus* biofilm formation and mature biofilms was previously reported [21]. In mature *S. aureus* biofilms, this mixture showed the potential to disassemble the biofilms at concentrations higher than 10 mM [21]. The antibiofilm activity of d-amino acids was also reported for other relevant pathogens, such as *Pseudomonas aeruginosa* and *S. epidermidis* [10]. Unfortunately, the mechanism of action of d-amino acids against biofilms is not completely known [21,40,41,42]. Regarding the MOX-loaded MLNs, both non-functionalized (MOX-MLNs) and functionalized (F-MOX-MLNs) nanoparticles showed a significant viability reduction in all strains at a MOX concentration of 0.5 µg mL^−1^, compared to the untreated control. At the highest concentration (2 µg mL^−1^), encapsulated and free MOX revealed biofilm viability of around 50% for ATCC 33591 and ATCC 6538 strains, and 25% for the *Xen36* strain.

For the three *S. aureus* strains studied, the crystal violet staining was used to quantify biofilm biomass after a 24 h treatment with MLNs or free MOX (Figure 4D–F). For MRSA, only MOX-loaded MLNs and free MOX at the highest concentration (2 µg mL^−1^) showed a significant reduction in the biofilm biomass compared to the untreated control. Interestingly, for the *S. aureus* ATCC 6538 strain, even at the lowest concentration tested (0.125 µg mL^−1^), MOX-loaded MLNs (MOX-MLNs and F-MOX-MLNs) and MOX-free showed a high effect on the biomass, with a reduction higher than 50%. Finally, for the *Xen36* strain, MOX-loaded MLNs and MOX-free showed a significantly high reduction at the two highest concentrations tested, while no effect on reducing biomass was observed for the unloaded MLNs.

Overall, the results from biofilm viability and biomass assays suggest that the MOX-loaded MLNs are more efficient against the methicillin-susceptible (MSSA) strains (ATCC 6538 and *Xen36*) than for the MRSA strain. The efficacy of MOX on mature MRSA biofilms has been previously reported [7]. However, the lower efficacy of MOX-loaded MLNs and free MOX for the MRSA strain may be a consequence of the different phenotype of the MRSA biofilms compared to the MSSA biofilms [43,44].

The antibiofilm efficiency of the NAC-loaded LNPs and the respective unloaded formulations was also evaluated using the XTT and crystal violet assays (Figure 5). For the three tested strains, all formulations showed a significant reduction of biofilm viability (higher than 50%) at the concentrations of 1 and 2 mg mL^−1^ of solid lipid. However, free NAC only showed a slight reduction of bacterial viability (lower than 25%) for the strain ATCC 6538. For the MRSA and *Xen36* strains, no significant decrease in biofilm viability was observed. These results were expected, since it is reported in the literature that even at a higher concentration (30 mM), NAC alone does not significantly reduce viability in MRSA biofilms [12]. The effect of the unloaded and NAC-loaded LNPs in biofilm biomass was also evaluated. All formulations showed a significant effect on bacterial biomass, which corroborates the results obtained for biofilm viability. Free NAC only at the highest concentration showed a significant reduction in biofilm biomass for the MRSA and *Xen36* strains.

Hence, the reduction of biofilm viability and biomass does not seem to be caused by the encapsulated NAC, but by the vehicle itself. The unloaded LNPs (non-functionalized and functionalized) previously showed a strong effect on the viability and biomass of *P. aeruginosa* biofilms [10]. This antibacterial effect is probably due to the use of Tween^®^80 as a surfactant in the production of the LNPs. This compound is well known for its antibacterial and antibiofilm properties against a wide range of bacterial strains, including *S. aureus* [45,46,47].

#### 3.4.2. Confocal Laser Scanning Microscopy (CLSM) Biofilm Analysis

Confocal laser scanning microscopy (CLSM) was used to assess the viability and integrity of the untreated and treated *Xen36* biofilms (Figure 6).

CLSM images revealed that untreated biofilms were dense and composed mainly of viable bacterial cells. The treatment with unloaded MLNs (non-functionalized and functionalized) showed no visible reduction of bacterial viability. The biofilms treated with F-MOX-MLNs or free MOX show a slightly higher number of dead cells in the deeper regions of the biofilm. Interestingly, the treatment with the functionalized MOX-loaded formulation reveals a less dense biofilm compared to free MOX. This result suggests a possible synergistic effect between the encapsulated antibiotic and the functionalization with d-amino acids. According to the literature, the combination of d-Phe, d-Pro, and d-Tyr not only prevented biofilm formation, but could also disassemble mature *S. aureus* biofilms [21]. In other studies, antibiotics in combination with d-amino acids revealed synergistic potential against *S. aureus* biofilms [22,48].

The treatment of biofilms with unloaded and NAC-loaded LNPs was also assessed by CLSM (Figure 6). Compared to the untreated control, all formulations and free NAC seem to have an effect by reducing the thickness of the biofilm. These findings are in accordance with the results previously obtained by the biofilm biomass study (Figure 5F). Hence, it is hypothesized that both unloaded and NAC-loaded LNPs promote disruption of the biofilm matrix, reducing the biofilm biomass. Additionally, functionalized LNPs (F-uLNPs and F-NAC-LNPs) seem to have an effect in reducing bacterial viability, since a higher number of dead cells is observable. It seems likely that this outcome may be a consequence of the functionalization rather than the presence of NAC. Nevertheless, non-functionalized and functionalized NAC-loaded LNPs previously reported a safer profile than the respective unloaded formulations [10].

Overall, CLSM biofilm analysis suggests that the MOX-loaded nanosystems have a bactericidal effect, while the NAC-loaded nanosystems seem to affect mainly the thickness of the biofilms. Thus, a combined nano therapy of the two nanosystems may be a valuable tool toward the eradication of *S. aureus* biofilms.

#### 3.4.3. Combined Nano Therapy of MOX-Loaded and NAC-Loaded Nanosystems

The potential of a nano therapy combining MOX-loaded (bactericidal) and NAC-loaded (matrix disruptive) nanosystems was evaluated using a more complex biofilm model, the in vitro catheter model. The biofilms were grown inside polyurethane catheter pieces for 24 h and further treated with the nanosystems alone or in combination for an additional 24 h.

Both unloaded and NAC-loaded LNPs at the solid lipid concentration of 2 mg mL^−1^, which corresponds to an NAC concentration of 0.9 mg mL^−1^, showed no effect on the biofilm viability for all the tested strains (Figure 7A). Likewise, free NAC at the same concentration did not significantly decrease biofilm viability after a 24 h treatment. These results differ from the previously observed data, obtained using the microtiter plate biofilm model (Figure 5A–C), where the LNP suspensions showed a biofilm viability reduction between 25 and 75%. Usually, in the microtiter plate model, the biofilm is loosely attached and can be easily detached during the washing steps. This model is also sensitive to sedimentation, where the biofilm forms mainly from the deposition of cells at the bottom of the wells [49]. Thus, it is possible that the different results from the two models are a consequence of the different characteristics of the formed biofilms and the limitations of the microtiter plate biofilm model.

The developed MLNs (unloaded and MOX-loaded) were also tested in the catheter biofilm model (Figure 7B). For the MRSA strain, only free MOX at a concentration of 0.5 µg mL^−1^ exhibited a significant decrease in bacterial viability compared to the untreated biofilms. Free MOX also showed a decrease in viability for the strain *Xen36*. These results suggest that not all encapsulated MOX is being released from the nanoparticles during the treatment period. For the *S. aureus* ATCC 6538, both free and encapsulated MOX did not affect the cell count. This outcome is probably related to the higher biofilm bacterial load observed in this strain, which is to be expected since this is considered a strong biofilm-forming strain [50].

In a more complex approach, the MOX- and NAC-loaded nanosystems were combined to assess potential synergistic effects (Figure 8). For the MRSA strain (Figure 8A,B), the combination of functionalized particles loaded with MOX (F-MOX-MLNs) and NAC (F-NAC-LNPs) exhibited the highest reduction in cell viability compared to the untreated control. Compared with F-MOX-MLNs alone, this combination showed a significantly lower viable count (*p* < 0.05), revealing a synergistic potential. The combination of F-MOX-MLNs and F-NAC-LNPs also revealed a significant effect on bacterial viability for the strain *Xen36*, compared to the untreated control (Figure 8E,F). However, the combination of F-MOX-MLNs with unloaded LNPs (uLNPs) and functionalized unloaded LNPs (F-uLNPs) presented even higher efficacy against *Xen36* biofilms. In a previously reported study, the unloaded non-functionalized and functionalized LNPs showed a higher potential against mature *P. aeruginosa* biofilms, compared to the NAC-loaded LNPs [10]. This outcome is probably associated with a higher exposition of the biofilms to the surfactant Tween^®^80 in the unloaded LNPs. Due to the presence of NAC in the loaded LNPs (NAC-LNPs and F-NAC-LNPs), it is believed that this compound is not only encapsulated but partially adsorbed at the surface of the nanoparticles, hindering the interactions between the Tween^®^80 and the biofilms. Despite the potential of these unloaded formulations, they previously revealed hemolytic activity against human red blood cells, which was not observed for NAC-loaded nanosuspensions [10]. Therefore, the combination of F-MOX-MLNs with F-NAC-LNPs is a safer therapeutic approach for future in vivo applications.

For the strain ATCC 6538, F-MOX-MLNs and free MOX showed the highest potential to reduce bacterial viability, in combination with the LNP suspensions (Figure 8C,D). The combination of free MOX with all LNP suspensions and free NAC showed a significant reduction of viability compared to the untreated control (*p* < 0.01). A similar effect was verified in the combination of F-MOX-MLNs and F-NAC-LNPs.

Overall, the combination of the functionalized, loaded nanosystems (F-MOX-MLNs and F-NAC-LNPs) showed a potential synergistic effect in the eradication of mature biofilms formed by the three *S. aureus* strains.

## 4. Conclusions

The formulations developed in this work exhibited low hydrodynamic sizes (below 300 nm), low PDI, and highly negative zeta potential values, which reveal a low tendency to form aggregates. The EE and LC were satisfactory, with no drug leakage observed over 12 weeks under storage conditions. The MLNs showed no cytotoxic effects on fibroblasts and no hemolytic activity against red blood cells at the tested concentrations, revealing a safe profile for further in vivo application.

The antibiofilm activity of both the MOX- and NAC-loaded nanosystems was assessed using the microtiter plate biofilm model in three *S. aureus* strains. Overall, the treatment with MOX-loaded MLNs led to a high reduction of biofilm biomass and viability in all strains tested. However, the formulations were more efficient against MSSA strains compared to MRSA. Both unloaded and NAC-loaded nanosystems also exhibited a significant reduction of biofilm viability and biomass, even at the lowest NAC concentration tested (0.45 mg mL^−1^). Hence, the presence of NAC does not seem to potentiate the antibiofilm effect verified for the vehicle itself. This outcome may be due to the use of Tween^®^80 as a surfactant, since it has reported antibacterial and antibiofilm activities against several bacterial strains, including *S. aureus* [45,46,51]. CLSM analysis of *Xen 36* biofilms after treatment with both MOX- and NAC-loaded nanosystems revealed that the latter affects mainly the thickness of the biofilm, while the treatment with MOX-loaded MLNs increases the population of dead bacterial cells within the structure.

In a more complex approach, the two nanosystems were combined to treat mature *S. aureus* biofilms grown in polyurethane catheters. For all the tested strains, the combination of functionalized nanoparticles encapsulating MOX (F-MOX-MLNs) and NAC (F-NAC-MLNs) revealed a potential synergistic effect in reducing the viable count within the biofilm. For the MRSA strain, the combination of these two formulations was the only condition where a significant reduction of the viable count was observed, compared to the untreated control. Thus, the combination of targeted nanosystems loading antibiotics and matrix-disruptive agents is a promising antibiofilm strategy to fight clinically relevant biofilms, such as *S. aureus* biofilms. Hence, the next steps to validate this strategy include assays with clinical isolates and in vivo experiments to assess the biodistribution and efficacy of the combinatory strategy using these developed nanosystems.

## Figures and Tables

**Figure 1 pharmaceutics-14-02294-f001:**
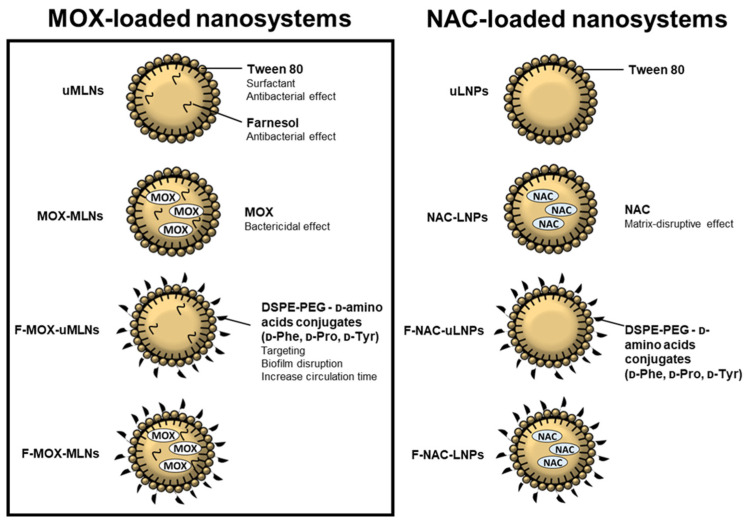
Schematic overview of the antibacterial and antibiofilm components of the nanosystems used for the proposed combinatory strategy. MOX-loaded MLNs were produced by an adapted protocol previously described by Cavalcanti et al. (2017) [23], while NAC-loaded nanosystems were produced by the double emulsion method [10]. The MLNs were composed of cetyl palmitate (solid lipid), Miglyol 812 (liquid lipid), farnesol, and the surfactants Tween 80 and Span 80. The LNPs were composed of cetyl palmitate and Tween 80. The surface of the nanoparticles was functionalized with DSPE-PEG-d-amino acid conjugates.

**Figure 2 pharmaceutics-14-02294-f002:**
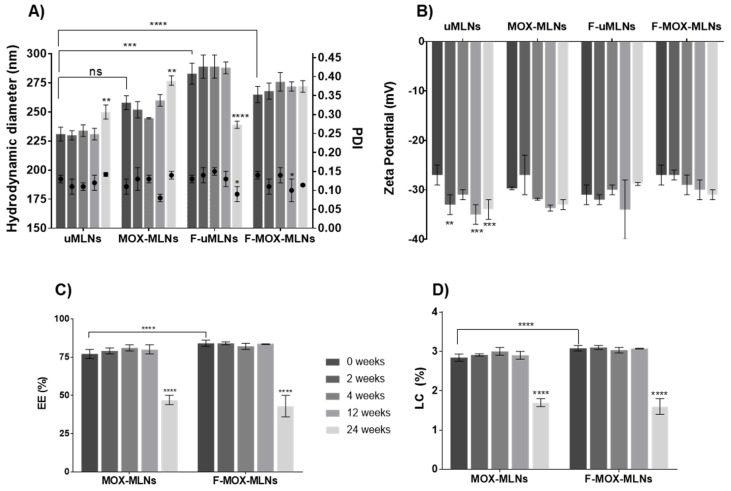
Characterization of the MLN suspensions over 24 weeks, stored at room temperature. (**A**) Hydrodynamic diameter and PDI. The bars represent the hydrodynamic diameter (left *y*-axis) and the dots represent the PDI (right *y*-axis). (**B**) Zeta potential. (**C**) EE. (**D**) LC. The parameters were evaluated at different time points (0, 2, 4, 12, and 24 weeks). The values are presented as the mean ± SD. ** *p* < 0.01, *** *p* < 0.001, relative to 0 weeks. * *p* < 0.05, ** *p* < 0.01, **** *p* < 0.0001. ns, not significant. Statistical analysis: two-way ANOVA, Tukey’s multiple comparisons test.

**Figure 3 pharmaceutics-14-02294-f003:**
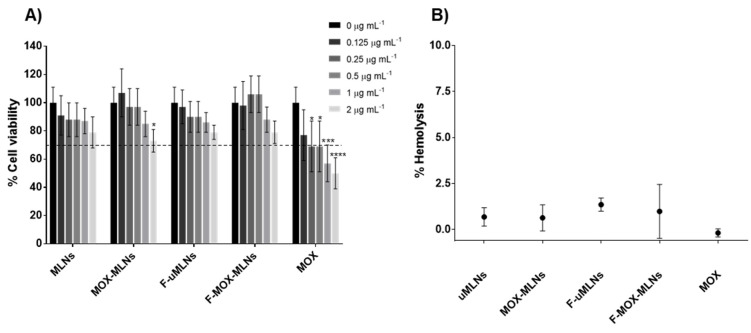
In vitro cytocompatibility studies. (**A**) MTT using the fibroblast L929 cell line. In this assay, different concentrations of encapsulated and free MOX were assessed (2.0, 1.0, 0.5, 0.25, and 0.125 µg mL^−1^). Triton^TM^-100x (1%, *v*/*v*) and DMEM were used as negative and positive controls, respectively. (**B**) Hemolysis assay. Encapsulated and free MOX at a concentration of 512 µg mL^−1^ were tested. For negative and positive controls, saline solution (0.85%, *w*/*v*) and Triton^TM^-100x (1%, *v*/*v*) were used, respectively. All values are presented as mean ± SD. * *p* < 0.05, *** *p* < 0.001, **** *p* < 0.0001, relative to the positive control. Statistical analysis: two-way ANOVA, Dunnett’s multiple comparisons test.

**Figure 4 pharmaceutics-14-02294-f004:**
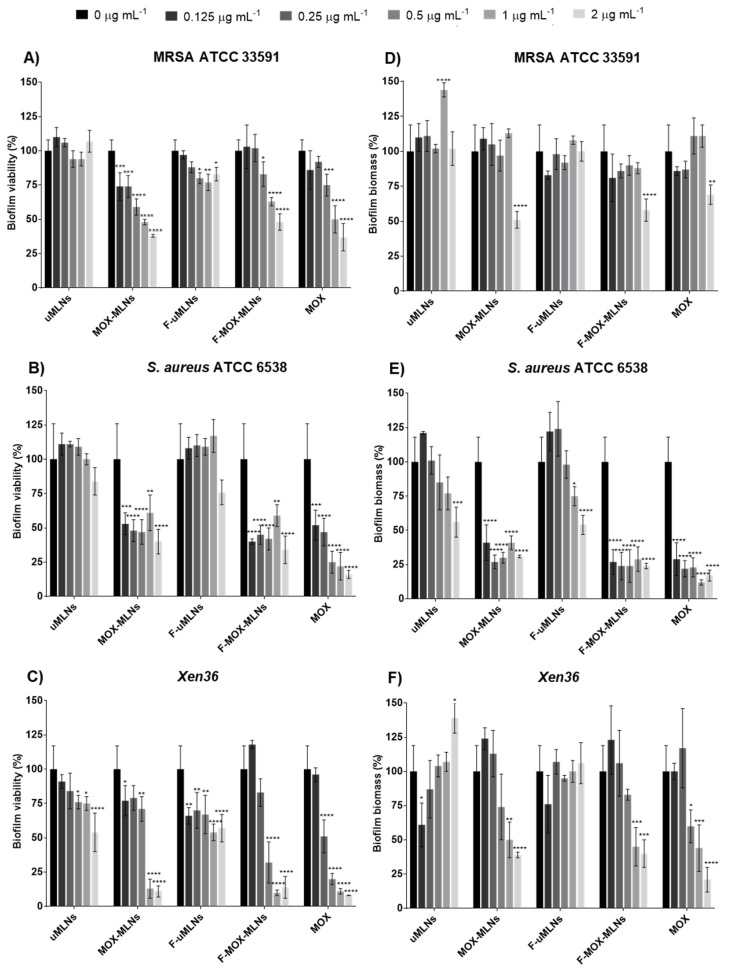
Quantification of (**A**–**C**) biofilm viability (XTT assay) and (**D**–**F**) biomass (crystal violet staining) after a 24 h treatment with MLNs and free MOX at the concentrations of 0, 0.125, 0.25, 0.5, 1, and 2 µg mL^−1^ of MOX. Unloaded MLNs were tested at the same concentration. The biofilms of (**A**,**D**) MRSA ATCC 33591, (**B**,**E**) *S. aureus* ATCC 6538, and (**C**,**F**) the bioluminescent strain *Xen36* were grown in 96-well plates for 24 h before the treatment. Untreated biofilms (0 µg mL^−1^ of MOX) were used as a positive control, and TSB 0.6x supplemented with 0.2% (*w*/*v*) glucose was used as a negative control. The values are represented as the mean ± SD. * *p* < 0.05, ** *p* < 0.01, *** *p* < 0.001, **** *p* < 0.0001 relative to 0 µg mL^−1^. Statistical analysis: two-way ANOVA, Dunnett’s multiple comparisons test.

**Figure 5 pharmaceutics-14-02294-f005:**
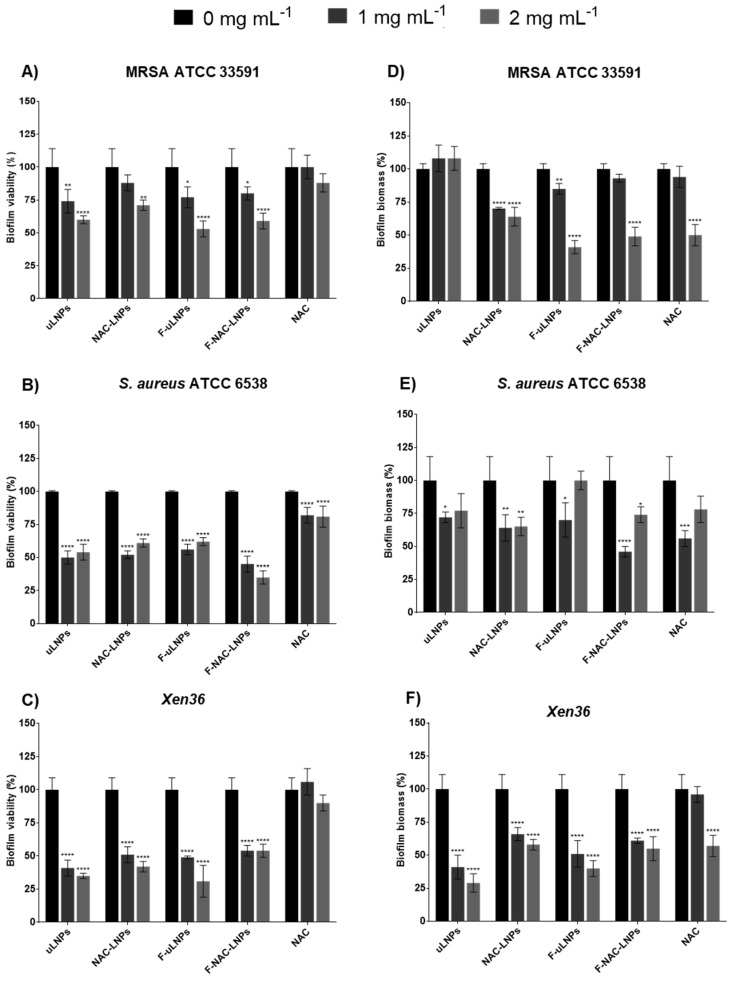
Quantification of (**A**–**C**) biofilm viability (XTT assay) and (**D**–**F**) biomass (crystal violet staining) after a 24 h treatment with LNP suspensions at the solid lipid concentrations of 0, 1, and 2 mg mL^−1^. Free NAC was tested at the concentrations of 0, 0.45, and 0.9 mg mL^−1^. The biofilms of (**A**,**D**) MRSA ATCC 33591, (**B**,**E**) *S. aureus* ATCC 6538, and (**C**,**F**) the bioluminescent strain *Xen36* were grown in 96-well plates for 24 h prior to the treatment. Untreated biofilms (0 mg mL^−1^ of solid lipid) were used as a positive control, and TSB 0.6x supplemented with 0.2% (*w*/*v*) glucose was used as a negative control. The values are presented as the mean ± SD. * *p* < 0.05, ** *p* < 0.01, *** *p* < 0.001, **** *p* < 0.0001 relative to 0 mg mL^−1^. Statistical analysis: two-way ANOVA, Dunnett’s multiple comparisons test.

**Figure 6 pharmaceutics-14-02294-f006:**
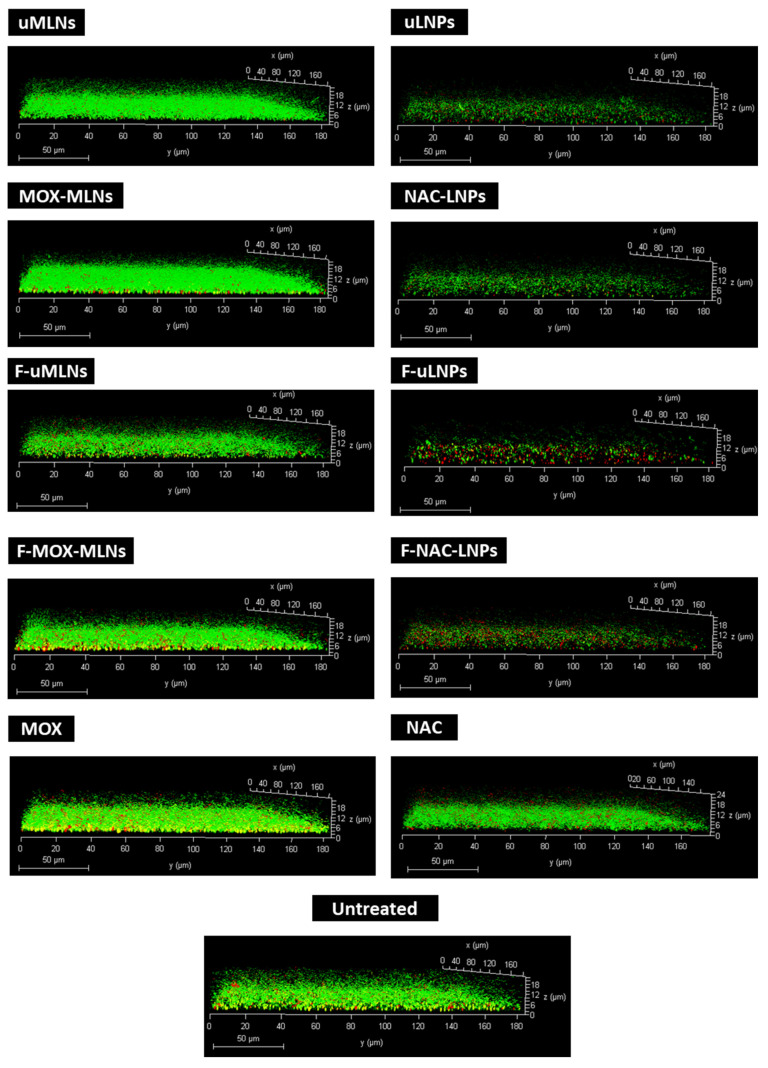
Three-dimensional CLSM images of untreated (control) and treated *Xen36* biofilms. The developed MLNs and free MOX were tested at a MOX concentration of 0.5 µg mL^−1^. For the biofilms treated with unloaded and NAC-loaded LNPs, a solid lipid concentration of 2 mg mL^−1^ (0.9 mg mL^−1^ of NAC) was used. Free NAC was used at the same concentration. The biofilms were stained with the LIVE/DEATH kit, where viable and dead cells are visualized in green and red, respectively. Untreated biofilms were used as a positive control. Scale bar, 50 µm.

**Figure 7 pharmaceutics-14-02294-f007:**
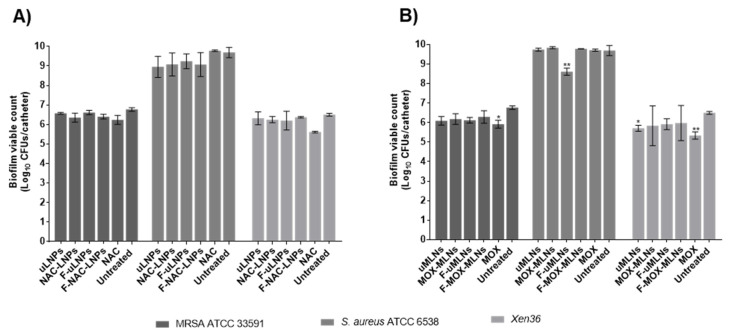
Biofilm viable count of the strains MRSA ATCC 33591, *S. aureus* ATCC 6538, and *Xen36* grown in vitro for 24 h in catheters, after a 24 h treatment with different nanosystems. (**A**) The unloaded and NAC-loaded LNPs were tested at a concentration of 2 mg mL^−1^ (corresponding to a NAC concentration of 0.9 mg mL^−1^). Free NAC was tested at the same concentration. (**B**) The developed MLNs and free MOX were tested at a MOX concentration of 0.5 µg mL^−1^. Untreated biofilms were used as a positive control, and TSB 0.6x supplemented with 0.2% (*w*/*v*) glucose was used as a negative control. The values are presented as the mean ± SD for three catheters. * *p* < 0.05, ** *p* < 0.01, relative to the untreated biofilms. Statistical analysis: two-way ANOVA, Dunnett’s multiple comparisons test.

**Figure 8 pharmaceutics-14-02294-f008:**
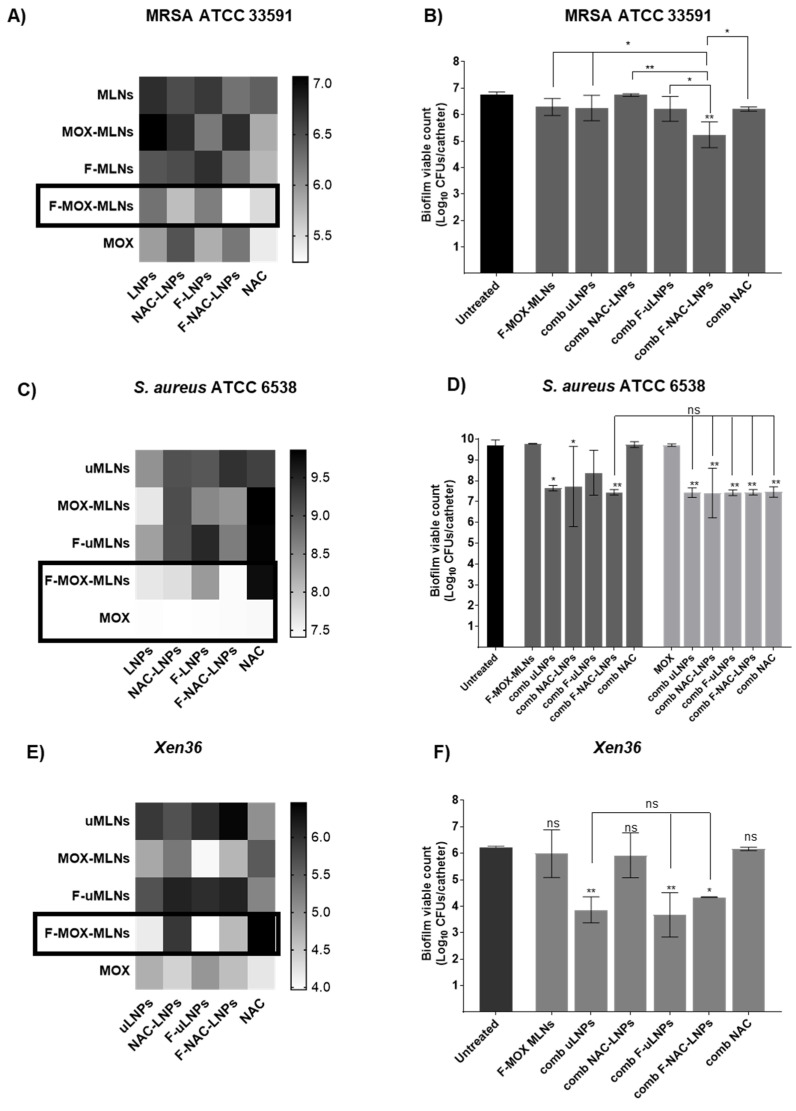
Biofilm viable count of (**A,B**) MRSA ATCC 33591, (**C,D**) *S. aureus* ATCC 6538, and (**E,F**) *Xen36* biofilms grown in vitro in catheters after treatment with the combined nano therapy. (**A,C,D**) Heat maps represent the effect of the combination of MOX- and NAC-loaded nanosystems on the biofilm viable count. The treatment combinations with lower viable counts are highlighted (black squares). (**B,E,F**) F-MOX-MLNs (dark grey) and free MOX (light grey) (0.5 µg mL^−1^) alone and in combination with NAC-loaded nanosystems (comb uLNPs, comb NAC-LNPs, comb F-uLNPs, and comb F-NAC-LNPs) or free NAC (comb NAC) at a solid lipid concentration of 2 mg mL^−1^ (corresponding to a NAC concentration of 0.9 mg mL^−1^). Untreated biofilms were used as a positive control, and TSB 0.6x supplemented with 0.2% (*w*/*v*) glucose was used as a negative control. The values are presented as the mean ± SD for three catheters. * *p* < 0.05, ** *p* < 0.01, relative to the untreated biofilms. ns, not significant; * *p* < 0.05, ** *p* < 0.01. Statistical analysis: one-way ANOVA, Tukey’s multiple comparisons test.

## Data Availability

Upon request, data will be provided.

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
