# Peer review of "Antibiofilm Combinatory Strategy: Moxifloxacin-Loaded Nanosystems and Encapsulated N-Acetyl-L-Cysteine"

_pharmaceutics, 2022, doi:10.3390/pharmaceutics14112294_

Round 1
Reviewer 1 Report
The reviewer thanks the authors for their exciting and encouraging paper. Nunes et al. focus on developing multiple lipid nanoparticles (MLNs) encapsulating the antibiotic moxifloxacin (MOX) to prevent Staphylococcus aureus-based bacterial biofilms. Literature studies are well presented, and illustrative drawings nicely demonstrate the idea. The article's topic will undoubtedly attract the interest of the scope audience of the journal. I would recommend this for publication without change in Pharmaceutics.

Author Response
We acknowledge the Reviewer for the overall appreciation of the manuscript.
Reviewer 2 Report
Authors developed multiple lipid nanoparticles (MLNs) to target and eradicate mature S. aureus biofilms. The study is worth noting and provides an interesting and promising approach to eradicating bacterial antibiofilm. The experimental approach and design are appropriate to adequately support the results that have been widely supported by the experimental evidence. The discussion and conclusion are well in line with the results.Author Response
We acknowledge the Reviewer for the overall appreciation of the manuscript.
Reviewer 3 Report
In the current submission, the authors have developed multiple lipid nanoparticles (MLNs) to encapsulate moxifloxacin. These nanoparticles were functionalized with á´…-amino acids to target the biofilm matrix. The overall finding of this manuscript is justified. In addition, an increased antibiofilm efficacy was observed when functionalized MOX-loaded MLNs were combined with NAC-loaded nanosystems. This work has several translational values. The following comments may be addressed during revision:
1. What about developed multiple lipid nanoparticles mono- or polydispersity?
2. In mtt assay, how the concentration of nanomaterials were determined?
3. In antibacterial assay, what were positive and negative controls?
4. I found minor issue, many typos and grammatical errors are seen in the paper. There are grammatical mistakes and typographical errors in the manuscript. The author should recheck this manuscript carefully and remove all such errors.
5. Future directions and future implications should be described in a clear manner with a strong conclusion.
6. A uniform presentation is required. The author should proofread the manuscript before the final submission.
Author Response
1. What about developed multiple lipid nanoparticles mono- or polydispersity?
As it can be seen in Figure 2A, all the formulations present a PDI below 0.15, during 24 weeks. In fact, we also refer that in the manuscript. Line 324-326: “The low PDI (<0.15) observed for all formulations suggests that they were monodisperse in size and did not tend to form aggregates over time, at the defined storage conditions”.
2. In mtt assay, how the concentration of nanomaterials were determined?
All the assays were performed as a function of the concentration of the encapsulated drug. Thus, the concentration of nanoparticles corresponds to the ones able to load different concentrations of MOX (2.0, 1.0, 0.5, 0.25, and 0.125 µg mL-1).
3. In antibacterial assay, what were positive and negative controls?
We are grateful for the Reviewer’ remark. In the antibacterial assay, the untreated bacterial suspension was used as a positive control and the MHB-Ca2+ medium was used as a negative control. This information was added to the manuscript, in the Section 2.7 of the Materials and Methods.
After corrections:
“Free NAC at the concentrations of 0.45 and 0.9 mg mL-1 (corresponding to solid lipid concentrations of 1 and 2 mg mL-1) was also tested. Untreated bacterial suspension was used as a positive control and MHB-Ca2+ medium was used as a negative control.”
After careful review of the manuscript, information regarding the positive and negative controls for the antibiofilm studies was also added to section 2.8 of the Materials and Methods.
After corrections:
“Untreated biofilms were used as a positive control and TSB 0.6x supplemented with 0.2% (w/v) glucose was used as a negative control.”
4. I found minor issue, many typos and grammatical errors are seen in the paper. There are grammatical mistakes and typographical errors in the manuscript. The author should recheck this manuscript carefully and remove all such errors.
We acknowledge the Reviewer for this comment. We have revised the whole manuscript and corrected several typos and grammatical mistakes.
5. Future directions and future implications should be described in a clear manner with a strong conclusion.
We have added to the conclusion the future directions of the work.
After corrections:
“Thus, the combination of targeted nanosystems loading antibiotics and matrix-disruptive agents is a promising antibiofilm strategy to fight clinically relevant biofilms, such as S. aureus biofilms. Hence, the next steps to validate this strategy include assays with clinical isolates and in vivo experiments to assess the biodistribution and efficacy of the combinatory strategy using these developed nanosystems.”
6. A uniform presentation is required. The author should proofread the manuscript before the final submission.
We have checked for all the incoherences and uniformized the whole manuscript.